# Agro-Industrial and Urban Compost as an Alternative of Inorganic Fertilizers in Traditional Rainfed Olive Grove under Mediterranean Conditions

**Laura L. de Sosa** [1,*], **Emilio Benítez** [2], **Ignacio Girón** [1] and **Engracia Madejón** [1]

[1] Instituto de Recursos Naturales y Agrobiología de Sevilla (IRNAS-CSIC), Av. Reina Mercedes 10, 41012 Sevilla, Spain; iggi@irnase.csic.es (I.G.); emadejon@irnase.csic.es (E.M.)

[2] Estación Experimental del Zaidin (EEZ-CSIC), Profesor Albareda 1, 18008 Granada, Spain; emilio.benitez@eez.csic.es

[*] Correspondence: lauralozano@irnsa.csic.es

**Abstract:** A three-year field study was conducted to evaluate the impact of two different agro-industrial byproducts on soil properties, provisioning services, olive quality and production in a traditional rainfed olive grove to assess suitable management options for recycling organic wastes and reduce the use of inorganic fertilizers. The organic amendments consisted of compost (AC), made from residues from the olive oil industry ("alperujo"), and biosolid compost (BC), constituted of wastewater sludge and green waste from parks and gardens. The compost addition enhanced carbon storage, available phosphorous and potassium content overtime, whereas no effect was detected on soil hydraulics, yield and olive trees growth, partly due to the high variability encountered among plots. Beneficial effects, especially carbon storage, were more evident during the fourth sampling, where carbon content increased by almost 40% for BC, suggesting that compost effects need to be evaluated in the long term. Strong seasonal changes of most of the physico-chemical parameters were detected, and therefore the effect of the compost could have been partly masked. Establishing a non-climatic variation scenario would be advisable to fully detect compost effects. Our results suggest that different agro-industrial byproducts could be potentially viable and valuable source of fertilization, favoring in this way a circular economy of zero waste.

**Keywords:** organic matter; ecosystem services; soil quality; productivity; quality of olive

## 1. Introduction

Agriculture in the Mediterranean area has been crucial for the economic development of its countries. In total, there are 7.7 million hectares devoted to olive crop only in the Mediterranean basin amounting to ca $11 \times 10^6$ ha around the world [1]. The olive grove sector is a major economic actor due to its ability to generate employment (employing more than 1,000,000 people per year), being the economic support of large areas of the Mediterranean basin and one of the sectors responsible for stopping rural depopulation [2]. Moreover, the olive tree plays a pivotal role in the maintenance of Mediterranean ecosystems. Therefore, the cultivation of the olive tree has become the backbone of the socio-economic and cultural life of many regions of the Mediterranean countries [1]. This fruit tree has been cultivated commercially for more than 4000 years, and, until recently, it was largely restricted to the Mediterranean region where it was grown in low-density plots (about 100 trees per hectare) and in low-rainfall areas [2]. However, during the 1990s, the production methods began to change rapidly. These changes were especially linked to an increase in density, a greater use of irrigation and an improvement in mechanization. This reconversion of olive groves has been more intense in those dedicated to oil than in those used for the table. The table olive grove sector, although less in cultivated area, is of great economic importance. Spain is the world's leading exporter of table olives, allocating

for this purpose on average 68% of production in recent years. In addition, the olive oil sector represents an important source of healthy food and is appreciated throughout the world as an integral element of the Mediterranean diet model, which since 2013 has been UNESCO's intangible cultural heritage and has become a hallmark of identity for some regions of the Mediterranean basin [3].

The Mediterranean countries share similar climatic and agro-ecological conditions, as well as two major limitations: deficit of water resources and the low level of soil organic matter. These two limitations make the entire basin very vulnerable to degradation and the advance of desertification. Around 74% of Mediterranean soils have an organic carbon content less than 2% [4]. This fact could seriously compromise the stability of soil functioning and, consequently, food security for the next decades.

In the Mediterranean areas, olive is often cultivated in shallow soils and marginal lands with traditional soil management techniques including frequent tillage and low organic matter inputs. These techniques associated to the limited soil coverage offered by the tree canopies further compromise the quality of these soils, increasing the risk of organic matter losses and desertification [5,6].

Some cultivation practices offer significant opportunities to maintain the functions associated with agricultural production, as well as to increase resilience to future disturbances. In this sense, FAO proposes the adoption of measures for the achievement of a "climate-smart agriculture", which pursues, among multiple objectives, the sustainable increase of productivity, carbon sequestration, the increase of the capacity of land recovery and the reduction of GHG emissions [7].

There is a growing recognition that soil organic matter is a key property of ecosystems to understand their stability in the face of global change [8]. The addition of exogenous organic matter to degraded soils can improve water retention capacity and hydraulic properties [9,10] making a more efficient water use under rainfed conditions. Organic amendments, if properly managed [11], are a pivotal factor in the provision of supporting ecosystem services due to its role in improving soil physical structure and the aggregates' stability in response to desiccation [12,13]. The influence of organic matter is not limited to its direct effects on soil physical and chemical processes but also affects the biotic communities [14]. Likewise, the amount of organic carbon can determine the stability of microbial communities against a series of disturbances [15].

The use of these organic amendments entails the activation of the biochemical cycles of nutrients with the consequent increase in their availability for crops. All these changes generally improve provisioning services, maximizing yield and production [16,17]. Specifically, organic amendments incorporated into olive orchards include compost, raw organic manure, olive mill wastewater, olive pomaces and chopped pruned material, among others. Direct application of non-composted material is less recommended. Composts made from different organic wastes are the best option for organic fertilization of olive crops [18]. Thus, the addition of organic amendments from organic wastes would provide a double solution: to address organic matter deficiency in systems with low inputs and comply with the obligation to manage this type of waste as established by European regulations.

The aim of this work was to evaluate the agricultural practices adopted (organic fertilization with two different composts) on soil properties including SOC and related functions (supply of nutrients, soil moisture content and hydraulic properties) and their effect on provisioning services focusing on production of table olives and quality of the olives.

## 2. Materials and Methods

### 2.1. Experimental Area and Experimental Design

The experiment site is representative of Mediterranean agriculture and located at the rainfed agriculture experimental farm "La Hampa" of the "Instituto de Recursos Naturales y Agrobiología de Sevilla (IRNAS-CSIC)". The soil is sandy clay loam soil, characterized by low fertility and low organic matter contents (pH: 7.5; TOC: 8 g kg$^{-1}$; N: 0.8 g kg$^{-1}$; Olsen P: 10 mg kg$^{-1}$; Available-K: 200 mg kg$^{-1}$). The climate is typically Mediterranean,

with mild rainy winters and very hot, dry summers. The environmental data obtained from the weather station at the experimental farm are summarized in Figure S1.

The experiment was established in 2018 in an area of 1.2 ha of olive groves of the "Manzanilla de Sevilla" variety with trees over 25 years old. The experimental area was divided into 20 plots of 140 m$^2$ (plantation frame 7 m $\times$ 5 m) integrated by nine olive trees with a single trunk and two main branches 0.8–1.5 m from the soil surface. The central tree was selected as the basis for soil sampling. A randomized block design with five treatments and four plots per treatment was selected for our study. The plantation has always remained rainfed. Before the establishment of the experiment, mineral fertilizer was applied as a basal dressing with 15-15-15 at a rate of about 100 kg ha$^{-1}$ adding 1 kg of urea per tree as cover fertilization. The crop has been supplemented in alternate years with foliar applications of $KNO_3$ and B before fruit set in organic and inorganic treatments.

Phytosanitary treatments consist of the application of Cu as a fungicide and two applications of dimethoate as an insecticide.

Two different organic amendments (at different doses) were applied in order to increase soil organic C content. The organic amendments consisted of compost (AC), made from residues from the olive oil industry mixed with citrus and legume residues ("alperujo") provided by the company FERTIORMONT Spain and biosolid compost (BC), provided by EMASESA, Sevilla, Southern Spain, constituted of wastewater sludge from a water treatment plant, and green waste from parks and gardens. The main characteristics of the two composts are shown in Table 1. The first application of compost was performed in February 2018 in the following doses: low dose (equivalent to 17.8 kg of AC or BC per tree around one meter of the base) and high dose (equivalent to 26.7 kg of AC+ or BC+ per tree around one meter of the base). Control plots without compost application and with the same mineral fertilization applied in previous years were also established each season. The second compost application was done in the same plots and at the same doses in December 2019 (Figure S1).

**Table 1.** Characterization of the compost used in this study.

| Parameter | AC (February 2018) | BC (February 2018) | AC (December 2019) | BC (December 2019) |
|---|---|---|---|---|
| Moisture | 23.4 $\pm$ 1.10 | 32.4 $\pm$ 1.31 | 18.1 $\pm$ 0.36 | 37.1 $\pm$ 1.23 |
| pH | 10.4 $\pm$ 0.06 | 6.82 $\pm$ 0.08 | 10.1 $\pm$ 0.07 | 6.41 $\pm$ 0.16 |
| CE (mS cm$^{-1}$) | 16.1 $\pm$ 0.55 | 7.03 $\pm$ 0.59 | 14.3 $\pm$ 0.47 | 7.82 $\pm$ 0.65 |
| OM (%) | 31.3 $\pm$ 0.75 | 32.8 $\pm$ 0.70 | 29.1 $\pm$ 0.45 | 31.1 $\pm$ 0.35 |
| N (%) | 0.68 $\pm$ 0.03 | 2.20 $\pm$ 0.003 | 0.63 $\pm$ 0.01 | 1.92 $\pm$ 0.01 |
| $P_2O_5$ (%) | 1.91 $\pm$ 0.05 | 3.12 $\pm$ 0.23 | 2.66 $\pm$ 0.09 | 2.84 $\pm$ 0.10 |
| $K_2O$ (%) | 4.87 $\pm$ 0.10 | 0.91 $\pm$ 0.006 | 9.58 $\pm$ 0.03 | 0.80 $\pm$ 0.01 |
| CaO (%) | 12.9 $\pm$ 0.32 | 7.12 $\pm$ 0.123 | 13.8 $\pm$ 0.32 | 4.50 $\pm$ 0.26 |
| Mg O (%) | 3.42 $\pm$ 0.05 | 1.97 $\pm$ 0.134 | 4.41 $\pm$ 0.22 | 2.21 $\pm$ 0.05 |
| Na (%) | 0.59 $\pm$ 0.03 | 0.51 $\pm$ 0.11 | 0.46 $\pm$ 0.02 | 0.55 $\pm$ 0.01 |
| $SO_3$ (%) | 0.55 $\pm$ 0.02 | 4.89 $\pm$ 0.14 | 0.66 $\pm$ 0.02 | 5.29 $\pm$ 0.16 |
| Fe (%) | 0.90 $\pm$ 0.03 | 1.71 $\pm$ 0.03 | 1.16 $\pm$ 0.02 | 3.33 $\pm$ 0.07 |
| Mn (mg kg$^{-1}$) | 254 $\pm$ 9.54 | 765 $\pm$ 51 | 312 $\pm$ 6.12 | 352 $\pm$ 12 |
| Cu (mg kg$^{-1}$) | 93.5 $\pm$ 3.80 | 210 $\pm$ 1.20 | 135 $\pm$ 3.76 | 153 $\pm$ 2.01 |
| Zn (mg kg$^{-1}$) | 69.0 $\pm$ 4.10 | 621 $\pm$ 5.9 | 78.0 $\pm$ 1.76 | 385 $\pm$ 12.3 |
| B (mg kg$^{-1}$) | 62.9 $\pm$ 3.76 | 39.9 $\pm$ 0.04 | 88.0 $\pm$ 2.40 | 20.0 $\pm$ 0.35 |
| As (mg kg$^{-1}$) | 1.72 $\pm$ 0.10 | 6.28 $\pm$ 0.73 | 0.91 $\pm$ 0.02 | 7.23 $\pm$ 3.68 |
| Cd (mg kg$^{-1}$) | 0.13 $\pm$ 0.00 | 0.74 $\pm$ 0.01 | 0.15 $\pm$ 0.02 | 1.08 $\pm$ 0.04 |
| Co (mg kg$^{-1}$) | 5.60 $\pm$ 0.23 | 31.4 $\pm$ 0.04 | 6.48 $\pm$ 0.42 | 7.90 $\pm$ 0.40 |
| Cr (mg kg$^{-1}$) | 42.7 $\pm$ 1.22 | 71.7 $\pm$ 0.06 | 55.8 $\pm$ 1.14 | 38.9 $\pm$ 1.48 |
| Ni (mg kg$^{-1}$) | 38.4 $\pm$ 1.70 | 33.8 $\pm$ 1.05 | 42.0 $\pm$ 2.03 | 20.8 $\pm$ 0.06 |
| Pb (mg kg$^{-1}$) | 13.7 $\pm$ 0.49 | 60.9 $\pm$ 0.41 | 9.23 $\pm$ 0.95 | 31.3 $\pm$ 2.76 |

### 2.2. Soil Sampling and Chemical Analysis

Four soil samplings around the central tree of each individual plot were taken at 0–15 cm in March 2018 (post first compost application), October 2018, October 2019 and May 2020 (post second compost application, Figure S1 and Table S1). The moist field soil was sieved (2 mm) and one sub-sample was stored at 4 °C prior to laboratory analysis while the other sub-sample was air dried, crushed and sieved (<2 mm and <60 μm) for chemical analysis.

Soil pH was measured in a 1 M KCl extract (1:2.5, *m/v*) after shaking for 1 h [19] using a pH meter (CRISON micro pH 2002). Electrical conductivity was determined in the water extract (1:5, *m/v*) after shaking for 1 h using Conductivity Meter (CRISON micro CM 2201). Total Organic Carbon (TOC) was calculated by dichromate oxidation and titration with ferrous ammonium sulphate [20]. Water-soluble carbon (WSC) content was determined using a TOC-VE Shimadzu analyzer after extraction with water using a sample-to-extractant ratio of 1:10.

Total Kjeldahl-N (TN) was determined by the method described by Hesse [19]. Available-P was determined after extraction with sodium bicarbonate at pH 8.5 [21], while available-K was determined after extraction with ammonium acetate at pH 7.5 [22]. Pseudo-total trace element concentrations in soil samples (<60 mm) of the last sampling (May 2020) were determined by digestion with aqua regia (1:3 v:v conc. $HNO_3$:HCl) in a microwave oven (Microwave Laboratory Station Mileston ETHOS 900, Milestone s.r.l., Sorisole, Italy) in all extracts were determined by ICP-OES using a Varian ICP720-ES (simultaneous ICP-OES with axially viewed plasma).

### 2.3. Soil Physical Properties

Soil hydraulic properties were measured "in situ" in January 2020 for the treatments of the highest doses of compost (AC+ and BC+) as well as for the control plots. The tension-disc infiltrometer method was used to determine in situ the hydraulic conductivity (K) and the sorptivity (S) in the range near saturation [23]. The measurements were carried out with a disc infiltrometer having a radius of 125 mm. A thin layer (2–3 mm depth) of fine sand, with a radius corresponding to that of the disc infiltrometer, was used to ensure a good contact between the membrane of the infiltrometer and the soil surface. The pressure potentials ($\psi_0$) chosen were −120, −80, −40 and −10 mm. The hydraulic conductivity, $K_0 = K(\psi_0)$, and the sorptivity, $S_0 = S(\psi_0)$, were obtained using the multi-disc approach described by Smettem and Clothier [23].

The bulk density of the soil was determined in undisturbed cores of 200 $cm^3$ volume. The initial volumetric water content ($\theta_n$) was determined from these soil cores. The volumetric water content, $\psi_0$ ($\theta_0$), was calculated from the water content of shallow samples scraped from the soil surface under the disc.

The gravimetric time defined as the time during which gravity controls the infiltration process was calculated as:

$$T_{grav} = (S_0/K_0)^2$$

where $S_0$ is the sorptivity and $K_0$ is the hydraulic conductivity determined as explained above.

Gravimetric water content (GWC) is the mass of water per mass of dry soil. It is measured by weighing a soil sample, drying the sample at 100 °C to remove the water and then weighing the dried soil.

### 2.4. Plant Development, Nutrition and Productivity

Crown volume was determined (October 2018, October 2019 and June 2020) by the measurement of the lengths of longest spread from edge to edge across the crown and the longest spread perpendicular to the first cross-section through the central mass of the crown [24].

At each plot, a representative sample of leaves of each tree was collected once a year (June 2018, June 2019 and June 2020) to identify nutrient deficiency or excess in plant tissue. Vegetal material (leaves) was washed with a 0.1 N HCl solution for 15 s and then with

distilled water for 10 s. Washed samples were oven dried at 70 °C. Dried plant material was ground and passed through a 500 μm stainless-steel sieve prior to preparation for analysis.

In each season, all treatments and were harvested the same day. The yield of each individual tree was weighted in the field. One sample per plot of around 1 kg was moved to the laboratory for the determination of several other properties related with the quality. Fruit size was estimated as the number of fruits per kilogram (USDA, 2019). Fruit load was estimated as the ratio between yield and fruit size in each plot. Ten fruits per plot were used in the measurements of fruit hardness per plot. Pulp hardness was measured with maximum peak force of the first compression [25] using a force gauge (FM 200, PCE Instruments, Spain). Maturity index [26] was used in 100 fruits per plot for the estimated change in fruit color. Pulp to stone ratio was measured in fresh and dry weight in 3 samples of ten fruits per plot.

### 2.5. Statistical Analysis

Statistical analysis was performed on the parameter values with four replicates of each treatment. Statistical analysis consisted of the assessment of the variance in the data, as explained in [27].

Analysis of variance (One-way ANOVA) followed by Tukey's post-hoc test was performed to explore the effect of compost application on soil physical and chemical properties within each year. All data were assessed for normality and homogeneity of variance with the Shapiro–Wilk test and Levene statistics, respectively. Log-transformations were made when necessary to meet the assumptions of normality and homoscedasticity. Pearson correlations were used to assess the relationship between variables. For all statistical tests, $p < 0.05$ was selected as the significance cut-off value. Statistical analysis was performed with SPSS v25 for Windows (IBM Corp., Armonk, NY, USA).

## 3. Results

### 3.1. Changes in Soil Properties

3.1.1. Changes in Soil Organic Matter

The compost addition had the most striking effect on C sequestration over time. Organic C accumulation followed a clear upward trend regardless of the treatment (Figure 1A).

However, this cumulative effect of C was more evident during the fourth sampling after the second compost addition, especially for the high dose of biosolid. The soil treated with both rates of AC compost increased the TOC by 20–25% with respect to control soils after the two applications of the product. Compost of biosolid had different quantitative effects depending on the doses: BC increased the TOC by 20% with respect to the control, whereas an increase of nearly 50% was observed in BC+ treatment.

Water-soluble carbon (WSC) was also positively influenced by compost addition (Figure 1B). The maximum values of this parameter were observed in the two first samplings, probably due to the higher soil moisture content, promoting the release of organic substances from fresh material during decomposition, thus contributing to soil nutrient cycling.

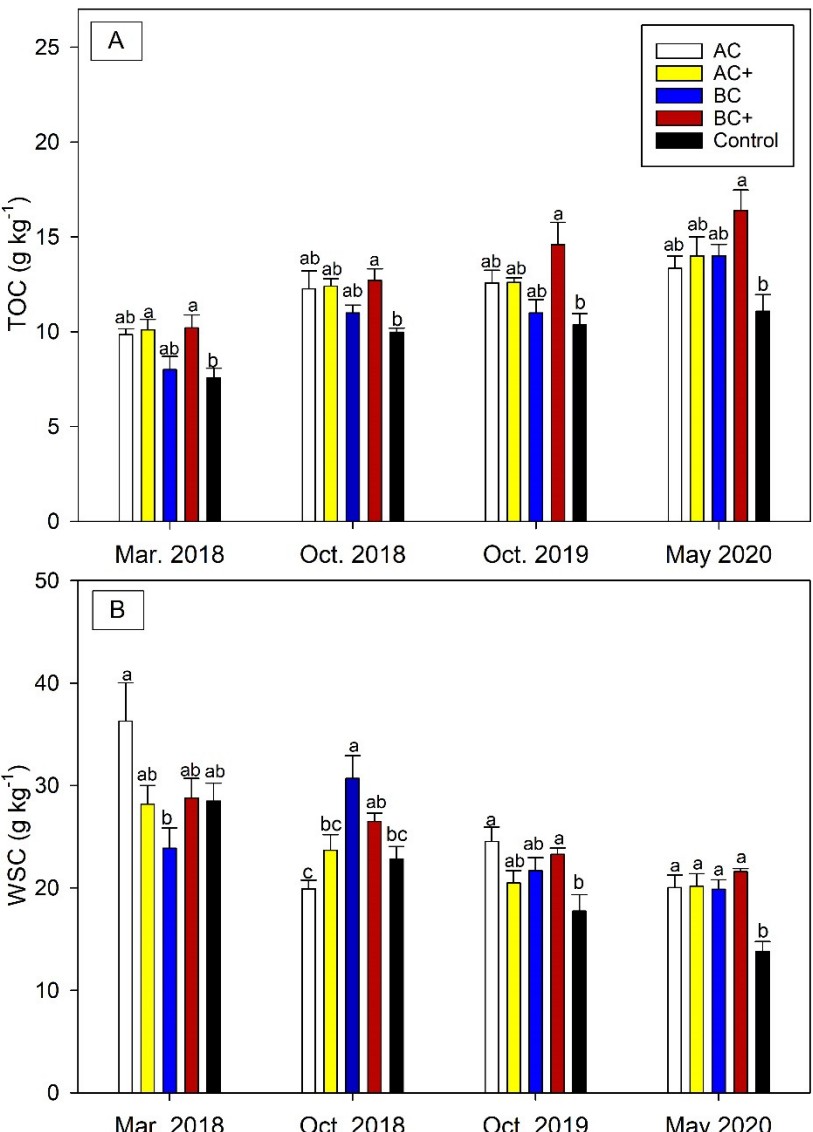

**Figure 1.** Evolution of (**A**) total organic carbon (TOC) and (**B**) water soluble Carbon (WSC) in soil during the 3 years of the experiment under different treatments: AC (Balperujo compost), AC+ (high dose of alperujo compost), BC (biosolid compost) and BC+ (high dose of biosolid compost). The same lowercase letters indicate no significant difference ($p > 0.05$) between treatments within each year. Data are mean values $\pm$ standard error of the mean (SEM).

### 3.1.2. Changes in Soil Fertility and Trace Element Contents

Soil physical and chemical parameters were strongly influenced by seasonal changes over time (Table 2).

**Table 2.** Soil parameters during the 3 years of experiments (see Figure S1) under different treatments: AC (alperujo compost), AC+ (high dose of alperujo compost), BC (biosolid compost) and BC+ (high dose of biosolid compost). The same lowercase letters indicate no significant difference ($p > 0.05$) between treatments within each year. Data are mean values $\pm$ standard error of the mean (SEM).

| | | AC | AC+ | BC | BC+ | Control |
|---|---|---|---|---|---|---|
| GWC (%) | March 2018 | 14.4 $\pm$ 0.50 | 14.5 $\pm$ 0.76 | 14.1 $\pm$ 0.57 | 15.8 $\pm$ 0.72 | 14.0 $\pm$ 0.48 |
| | October 2018 | 11.5 $\pm$ 0.68 | 11.8 $\pm$ 0.07 | 12.4 $\pm$ 0.60 | 13.7 $\pm$ 0.73 | 12.3 $\pm$ 0.30 |
| | October 2019 | 2.45 $\pm$ 0.66 | 1.91 $\pm$ 0.76 | 2.69 $\pm$ 0.93 | 1.73 $\pm$ 0.17 | 2.96 $\pm$ 0.31 |
| | May 2020 | 5.47 $\pm$ 0.39 | 3.67 $\pm$ 0.54 | 4.67 $\pm$ 0.68 | 4.16 $\pm$ 0.50 | 3.44 $\pm$ 0.70 |
| pH | March 2018 | 7.81 $\pm$ 0.09 | 7.59 $\pm$ 0.06 | 7.60 $\pm$ 0.11 | 7.50 $\pm$ 0.07 | 7.51 $\pm$ 0.09 |
| | October 2018 | 7.89 $\pm$ 0.09 | 7.66 $\pm$ 0.06 | 7.68 $\pm$ 0.11 | 7.57 $\pm$ 0.07 | 7.59 $\pm$ 0.09 |
| | October 2019 | 7.32 $\pm$ 0.03 | 7.33 $\pm$ 0.06 | 7.44 $\pm$ 0.05 | 7.31 $\pm$ 0.04 | 7.37 $\pm$ 0.05 |
| | May 2020 | 8.62 $\pm$ 0.04 [a] | 8.59 $\pm$ 0.04 [a] | 7.28 $\pm$ 0.27 [b] | 7.33 $\pm$ 0.17 [b] | 7.18 $\pm$ 0.42 [b] |
| EC (mS cm$^{-1}$) | March 2018 | 208 $\pm$ 37 | 139 $\pm$ 26 | 118 $\pm$ 25 | 137 $\pm$ 22 | 128 $\pm$ 22 |
| | October 2018 | 199 $\pm$ 35 | 133 $\pm$ 25 | 112 $\pm$ 24 | 131 $\pm$ 21 | 122 $\pm$ 21 |
| | October 2019 | 64 $\pm$ 6.1 | 58 $\pm$ 7.7 | 54 $\pm$ 8.0 | 59 $\pm$ 4.5 | 51 $\pm$ 9.7 |
| | May 2020 | 194 $\pm$ 20 | 203 $\pm$ 13 | 145 $\pm$ 35 | 137 $\pm$ 17 | 116 $\pm$ 21 |
| Kjeldalh-N (g kg$^{-1}$) | March 2018 | 0.88 $\pm$ 0.03 | 0.94 $\pm$ 0.07 | 0.95 $\pm$ 0.03 | 0.94 $\pm$ 0.01 | 0.85 $\pm$ 0.02 |
| | October 2018 | 0.89 $\pm$ 0.05 | 1.06 $\pm$ 0.13 | 1.03 $\pm$ 0.04 | 0.86 $\pm$ 0.04 | 0.97 $\pm$ 0.03 |
| | October 2019 | 1.04 $\pm$ 0.10 | 0.95 $\pm$ 0.08 | 0.93 $\pm$ 0.09 | 1.19 $\pm$ 0.10 | 0.91 $\pm$ 0.12 |
| | May 2020 | 1.04 $\pm$ 0.09 | 1.04 $\pm$ 0.06 | 1.10 $\pm$ 0.11 | 1.35 $\pm$ 0.15 | 1.05 $\pm$ 0.08 |
| Olsen-P (mg kg$^{-1}$) | March 2018 | 10.4 $\pm$ 0.74 | 15.7 $\pm$ 1.59 | 13.7 $\pm$ 1.37 | 12.7 $\pm$ 2.15 | 10.2 $\pm$ 1.48 |
| | October 2018 | 10.0 $\pm$ 0.79 | 16.2 $\pm$ 1.60 | 14.9 $\pm$ 2.83 | 13.2 $\pm$ 1.31 | 11.7 $\pm$ 1.04 |
| | October 2019 | 15.7 $\pm$ 2.28 [a,b] | 14.0 $\pm$ 1.78 [b] | 16.2 $\pm$ 1.86 [a,b] | 22.2 $\pm$ 1.69 [a] | 12.6 $\pm$ 1.02 [b] |
| | May 2020 | 26.7 $\pm$ 3.12 [b] | 25.6 $\pm$ 0.93 [b] | 33.8 $\pm$ 2.14 [b] | 47.6 $\pm$ 4.88 [a] | 23.5 $\pm$ 1.25 [b] |
| Available-K (mg kg$^{-1}$) | March 2018 | 278 $\pm$ 7.4 | 325 $\pm$ 45 | 275 $\pm$ 58 | 283 $\pm$ 40 | 208 $\pm$ 9.4 |
| | October 2018 | 269 $\pm$ 34 [a,b] | 448 $\pm$ 79 [a] | 305 $\pm$ 19 [a,b] | 333 $\pm$ 55 [a,b] | 230 $\pm$ 13 [b] |
| | October 2019 | 267 $\pm$ 19 [a,b] | 302 $\pm$ 18 [a] | 167 $\pm$ 20 [c] | 209 $\pm$ 15 [b,c] | 157 $\pm$ 26 [c] |
| | May 2020 | 840 $\pm$ 132 [a] | 893 $\pm$ 94 [a] | 245 $\pm$ 27 [b] | 383 $\pm$ 46 [b] | 296 $\pm$ 28 [b] |

Focusing on the treatments effect, an upward trend was observed in pH in AC treatments over time, while, for BC treatments and the control, the pH showed a slight decrease in these values. In the sampling of October 2019, significant drops in pH values and in moisture content for all treatments were identified, probably caused by the scarce rainfall of that year. Values of pH were maintained close to neutrality in all treatments in the first three samplings without the addition of compost causing significant changes (Table 2). However, the second compost addition (December 2019) caused an increase in pH values in the AC treatments by more than one point with respect to the rest of the treatments that maintained values similar to those obtained in previous samplings. When compost is added to soils, one of the risks is to increase their salinity. However, in this trial, after two additions of compost, the EC values remained low without observing significant differences between the organic and the inorganic treatments (Table 2).

Fertilizer value of the products tested was evaluated by the measurement of N-Kjeldalh, P-Olsen and available K (Table 2). In general, in the two first samplings, values of these parameters were similar between treatments and no statistical differences were found. However, in the two last samplings, some differences were observed. Fertility in terms of P (particularly in BC treatments) and K (particularly in AC treatments) was enhanced. The BC+ treatment also presented the highest values in terms of P accumulation and N in the last sampling compared to the rest of the treatments. With respect to K, the accumulation in the AC and AC+ treatments stood out for the last sampling, where there was an increase of approximately 170% with respect to the control.

The concentration of trace elements in the soil was determined in the last of the samplings carried out (Table S2). Overall, the concentration of metals and trace elements in soils treated with compost remained very similar to those obtained for the control.

### 3.1.3. Changes in Soil Hydraulic and Physical Properties

Gravimetric water content (GWC) in soil was more influenced by seasonal weather conditions than compost addition (Table 2 and Figure S1). In the two first samplings, values were higher than those obtained in the two following samplings because the rainfall occurred before the samplings (Figure S1). Although there were no significant differences between treatments, values of GWC in amended soils (especially with the high doses of BC) tended to be slightly higher than in control soils.

The effect of the compost addition at the high doses (AC+ and BC+) was studied on some key physical properties such as hydraulic sorptivity (S), gravimetric time (T grav), hydraulic conductivity (K) and frame-weighted mean pore size ($\lambda$m) at different pressure potentials (Figure 2). Overall, the effect of the high doses of both composts did not have a significant impact on soil physical properties, partly due to the high variability encountered among plots. However, we could appreciate some differences with respect to the control. Sorptivity was almost two times greater in control than in soils treated at maximum pressure potential (−120), whereas $\lambda$m differed approximately two-fold in amended soils with respect to the control.

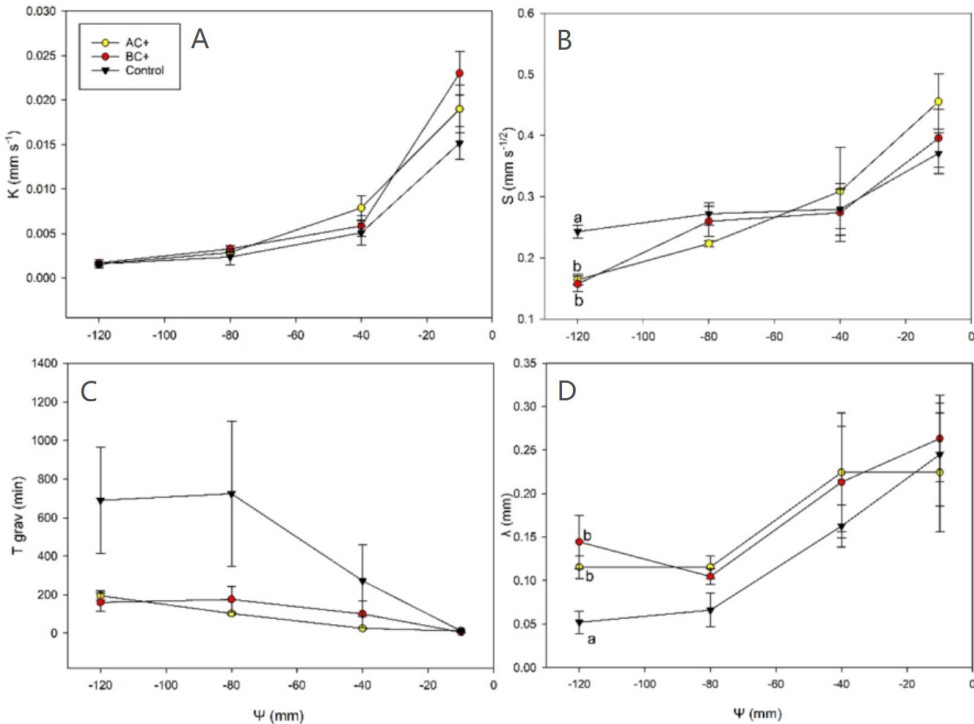

**Figure 2.** (**A**) Hydraulic conductivity (K), (**B**) sorptivity (S), (**C**) gravimetric time (T grav) and (**D**) frame-weighted mean pore size ($\lambda$m) at different potential under different treatments: AC (alperujo compost), AC+ (high dose of alperujo compost), BC (biosolid compost), BC+ (high dose of biosolid compost). The same lowercase letters indicate no significant difference ($p > 0.05$) between treatments within each year. Data are mean values ± standard error of the mean (SEM).

### 3.2. Changes in Crop

### 3.2.1. Changes in Crop Development and Nutritional Status in Leaves

Tree crown size was initially characterized by very similar values, although the individuals corresponding to BC treatments tended to have a slightly lower size (Figure 3A).

The increase in canopy volume did not provide a clear indication of the positive effects of the organic amendments on vegetative activity.

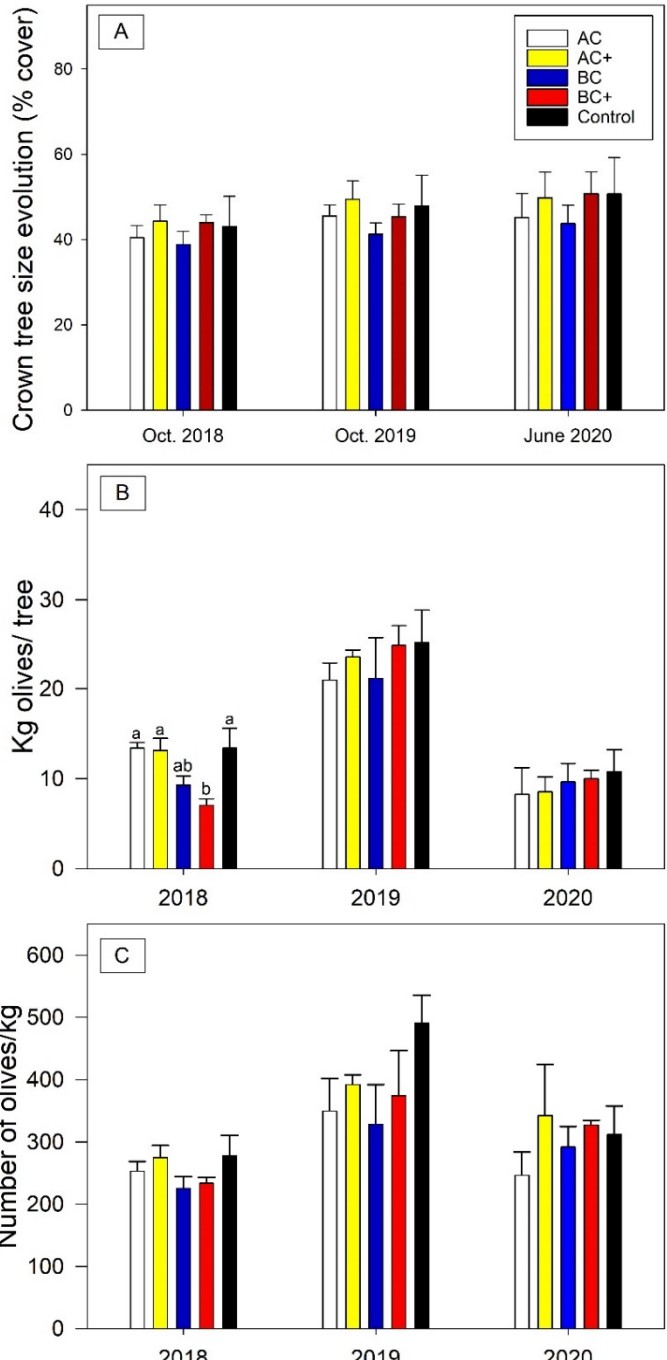

**Figure 3.** (**A**) Crown tree size evolution; (**B**) olive production (kg/ tree); and (**C**) relation of number of olives per kg during the 3 years of the experiment under different treatments: AC (alperujo compost), AC+ (high dose of alperujo compost), BC (biosolid compost) and BC+ (high dose of biosolid compost). The same lowercase letters indicate no significant difference ($p > 0.05$) between treatments within each year. Data are mean values $\pm$ standard error of the mean (SEM).

Except for N, leaf nutritional levels were generally within the appropriate limits for olive tree crop (Table 3) according to the reference parameters offered by Fernández-Escobar [28] elaborated from data collected by Chapman [29], Childers [30] and Beutel et al. [31]. Regarding P and K content in leaves, a general decrease was observed in the second year regardless

of the treatment (Table 3). Nevertheless, the compost seemed to have an interesting fertilizer effect for both nutrients.

**Table 3.** Macronutrients in leaf during the 3 years of experiments under different treatments: AC (alperujo compost), AC+ (high dose of alperujo compost), BC (biosolid compost) and BC+ (high dose of biosolid compost). Data are mean values ± standard error of the mean (SEM).

| | | AC | AC+ | BC | BC+ | Control |
|---|---|---|---|---|---|---|
| N (%) | 2018 | 1.12 ± 0.06 | 1.09 ± 0.05 | 1.04 ± 0.05 | 1.16 ± 0.04 | 1.17 ± 0.04 |
| | 2019 | 0.86 ± 0.04 | 0.87 ± 0.03 | 0.79 ± 0.05 | 0.87 ± 0.09 | 0.93 ± 0.07 |
| | 2020 | 1.23 ± 0.02 | 1.26 ± 0.02 | 1.24 ± 0.03 | 1.3 ± 0.04 | 1.32 ± 0.05 |
| P (%) | 2018 | 0.10 ± 0.005 | 0.10 ± 0.004 | 0.09 ± 0.003 | 0.10 ± 0.004 | 0.10 ± 0.005 |
| | 2019 | 0.07 ± 0.01 | 0.06 ± 0.002 | 0.05 ± 0.004 | 0.06 ± 0.01 | 0.07 ± 0.004 |
| | 2020 | 0.14 ± 0.01 | 0.13 ± 0.003 | 0.12 ± 0.01 | 0.14 ± 0.001 | 0.13 ± 0.01 |
| Available-K (%) | 2018 | 0.96 ± 0.02 | 0.94 ± 0.01 | 0.91 ± 0.02 | 0.94 ± 0.04 | 0.92 ± 0.02 |
| | 2019 | 0.53 ± 0.05 | 0.55 ± 0.05 | 0.47 ± 0.05 | 0.52 ± 0.09 | 0.48 ± 0.04 |
| | 2020 | 1.12 ± 0.06 | 1.16 ± 0.02 | 1.02 ± 0.01 | 1.03 ± 0.05 | 1.02 ± 0.02 |
| Ca (%) | 2018 | 0.71 ± 0.05 | 0.71 ± 0.05 | 0.71 ± 0.02 | 0.74 ± 0.04 | 0.76 ± 0.01 |
| | 2019 | 1.75 ± 0.15 | 1.69 ± 0.11 | 1.56 ± 0.19 | 1.53 ± 0.14 | 1.73 ± 0.09 |
| | 2020 | 1.12 ± 0.10 | 1.01 ± 0.06 | 1.15 ± 0.21 | 1.21 ± 0.13 | 1.06 ± 0.08 |
| Mg (%) | 2018 | 0.08 ± 0.004 | 0.08 ± 0.01 | 0.07 ± 0.01 | 0.08 ± 0.002 | 0.08 ± 0.004 |
| | 2019 | 0.10 ± 0.01 | 0.09 ± 0.01 | 0.09 ± 0.01 | 0.09 ± 0.01 | 0.10 ± 0.005 |
| | 2020 | 0.08 ± 0.01 | 0.07 ± 0.004 | 0.07 ± 0.01 | 0.08 ± 0.01 | 0.08 ± 0.004 |
| Na (%) | 2018 | 0.02 ± 0.003 | 0.02 ± 0.01 | 0.04 ± 0.01 | 0.03 ± 0.01 | 0.02 ± 0.005 |
| | 2019 | 0.14 ± 0.02 | 0.13 ± 0.01 | 0.17 ± 0.01 | 0.17 ± 0.03 | 0.16 ± 0.02 |
| | 2020 | 0.04 ± 0.01 | 0.03 ± 0.002 | 0.05 ± 0.01 | 0.05 ± 0.01 | 0.04 ± 0.003 |

Trace element accumulation in the leaves revealed no significant changes in any treatment with respect to the control (Table S3). However, the results reflect a marked seasonality in trace elements availability between sampling times.

### 3.2.2. Changes in Crop Productivity and Harvest Quality

In general, despite initial low levels of fertility (with <1% of TOC content), soil in this study was able to sustain the olive crops tested with acceptable levels of productivity along the campaigns preceding this experimentation. No significant changes in fruit production between AC treatment and the mineral control were observed in the first post-application year (October 2018, Table S1) (Figure 3B).

In the second year, a remarkable increase of the yield was observed irrespective of the treatment. In this second season, production obtained with the high doses of both composts were similar to that obtained in mineral control plots.

Although not significant, larger olives were obtained according to fruits per kg (quality data for table olives) in the organic treatments in the two first seasons (Figure 3C). In the third and last year of collection, marked by a new drop in production, the size of the fruit was similar in all treatments except for the olives obtained with the low dose of biosolids, which were clearly the largest.

Other parameters related with the olive quality are presented in Table 4. In the first season, based on the data of maturity index (MI), pulp stone weight ratio fresh (P/S f) and dry (PS d) and hardness (H), the quality of the olives was slightly higher in control mineral treatment, although the differences were not significant for any of the parameter evaluated. In the two following seasons, the parameters describing main quality factors for table olives were similar in all treatments

**Table 4.** Summary of yield quality during the 3 years of the experiment under different treatments: AC (alperujo compost), AC+ (high dose of alperujo compost), BC (biosolid compost) and BC+ (high dose of biosolid compost). Data are mean values $\pm$ standard error of the mean (SEM).

|  |  | AC | AC+ | BC | BC+ | Control |
|---|---|---|---|---|---|---|
| MI | 2018 | 0.84 ± 0.06 | 1.15 ± 0.29 | 1.11 ± 0.18 | 1.38 ± 0.28 | 0.70 ± 0.11 |
|  | 2019 | 1.06 ± 0.27 | 0.81 ± 0.33 | 0.99 ± 0.31 | 1.13 ± 0.23 | 1.23 ± 0.29 |
|  | 2020 | 1.64 ± 0.28 | 1.42 ± 0.13 | 1.38 ± 0.43 | 2.42 ± 0.32 | 1.63 ± 0.40 |
| P/S f | 2018 | 4.22 ± 0.31 | 4.54 ± 0.38 | 4.36 ± 0.23 | 3.76 ± 0.47 | 4.38 ± 0.47 |
|  | 2019 | 6.46 ± 0.23 | 5.11 ± 0.47 | 6.04 ± 0.72 | 4.95 ± 0.50 | 5.42 ± 0.63 |
|  | 2020 | 5.67 ± 0.63 | 5.71 ± 0.35 | 7.15 ± 0.31 | 6.68 ± 1.14 | 6.01 ± 1.02 |
| P/S d | 2018 | 2.48 ± 0.11 | 2.55 ± 0.11 | 2.48 ± 0.11 | 2.07 ± 0.10 | 2.58 ± 0.21 |
|  | 2019 | 6.11 ± 0.51 | 5.03 ± 0.62 | 5.24 ± 0.34 | 4.54 ± 0.29 | 5.51 ± 0.58 |
|  | 2020 | 4.10 ± 0.39 | 3.93 ± 0.35 | 4.34 ± 0.51 | 4.36 ± 0.20 | 4.21 ± 0.41 |
| H | 2018 | 43.7 ± 1.07 | 44.1 ± 5.30 | 45.5 ± 3.19 | 40.8 ± 1.14 | 46.5 ± 1.46 |
|  | 2019 | 48.1 ± 2.25 | 49.1 ± 1.98 | 44.4 ± 2.83 | 48.3 ± 2.54 | 50.1 ± 1.27 |
|  | 2020 | 54.6 ± 5.91 | 52.9 ± 1.48 | 50.6 ± 2.81 | 52.2 ± 2.62 | 52.1 ± 0.38 |

Maturity Index (MI); pulp stone weight ratio, fresh (P/S f) and dry (P/S d); hardness. (H).

## 4. Discussion

### 4.1. Potential Positive and Neutral Effects of Compost Addition

Numerous benefits of the compost addition from an agronomic point of view have been identified in previous studies [32,33]. These potential benefits include the improvement of nutrient supply and C sequestration, crop quality and yield and soil moisture, among other major findings. Specifically, some authors have observed an increased in organic matter in soils amended with compost from olive residues [34] or with biosolid compost [35], which is in good agreement with findings in this study.

The increase of C content in soils has been directly linked with an improvement in soil quality and ecosystem functioning. However, enhancing C storage without compromising sustainability and profitability requires innovative solutions [36–38]. Both types of compost proposed in this work ("alperujo and biosolid") are generated in large quantities during short periods of time, thus its use and recycling represent a major challenge but also an excellent opportunity to increase carbon stocks in the soil [34,39]. Thus, the addition of this type of compost could offer a suitable management option to add value to this byproduct, enhance the sustainability of the olive oil production system, increase farm productivity favoring food security and reduce the effects of climate change [13,40]. However, further studies should be undertaken to explore the dynamics of this C in the soil.

A similar pattern as per TOC could be seen for WSC in soils. The water-soluble C fractions constitute the easily decomposable part of the TOC, being a source of energy for microorganisms [41]. Our results suggest that different mineralization rates were taking place over the years. At the beginning of the experiment, differences in WSC values with respect to the control were barely noticeable until microorganism adapted to the exogenous input of OM. At the end of the experiment, the difference in available C for all treatments was much more obvious with respect to the control. This behavior was also observed by Madejón et al. [17], with similar amendments in degraded soils under fast-growing trees.

Regarding soil fertility and content of trace elements, strong seasonal patterns rather than compost effects were detected. However, an increase in pH was detected in the last sampling for AC treatment. The AC compost had a fairly alkaline pH (Table 1), which may have caused this increase in the soils. It is necessary to monitor this effect since the availability of nutrients and micronutrients for the plant can be affected by the increase in the alkalinity of the soils [42].

Fertility in terms of increasing P and K soil content, especially in the last two samplings, was also a valuable factor of the compost addition worth stressing. The results obtained in this study agree with those obtained by Ciadamidaro et al., [43] using the same type of compost, showing that both fertilizers turned out to be an important source of P and K for the crops. In contrast, we did not observe any significant increment in N soil content, even though the biosolid compost was relatively enriched in N. This lack of effect contrasts with studies such as that of Fernández-Hernández et al. [34], where a significant increment in N content after six years was identified. The fact that they added additional sources of N to the compost might be a possible explanation.

When biosolids compost is used, there is a risk of increasing the content of trace elements in the soils and consequently increasing their phytoxicity [44]. However, in the present experiment, the concentrations of metals and trace elements in soils treated with BC, even in the case of high doses, were comparable to those treated with AC compost and the control. These values were much lower than those considered toxic for soils [45] (Table S2). These results suggest that, when a well stabilized compost is applied, the risks of phytotoxicity and adverse effect on the environment decreases [16,46].

Several studies have also indicated the potential use of organic compost to improve soil physical parameters and therefore soil workability. Our results do not suggest a massive impact of the compost addition on soil hydraulic conditions, but some improvements were observed. These results contrast with studies such as that of Tsadilas et al. [47], where a positive response in soils regarding water retention capacity and availability after 3 years of biosolid application was shown. However, in our study, the scarce and irregular rainfall could have masked this effect.

Although not significant, the compost addition (especially BC+) tended to slightly improve the soil hydraulic conditions, as has been shown before [10,48]. In our study, a decrease in the mean pore size was observed in general with respect to the suction pressure in all treatments (Figure 2D), although in the control soils this decrease seemed more pronounced. This fact was closely related to the lower sorptivity values in the organic amended soils at −120 mm, probably due to the addition of larger particle sizes included in the compost, modifying therefore soil texture [49]. Regarding the gravimetric time (T grav), it was difficult to establish a clear impact of the compost addition on this soil physical property due to the great variability of the untreated soils. Overall, it seems to indicate that the infiltration time was more controlled by sorptivity in control mineral soils, where the smaller pore size probably gave place to poorer hydraulic conductivity, whereas the process is mainly controlled by gravity in amended soils.

Although these were not statistically significant effects, the trend of improvement in soil physical properties was confirmed, especially in the case of treatment with biosolids compost. Our results agree with those of González et al. [50], who reported the impact of composted sewage sludge on soil physical properties, which affected infiltration and erosion. In most cases, the changes in bulk density, cracking, direct measurements of infiltration, water retention capacity, percentage of useful water and simulator erosion measurements were higher in amended soil but not sufficiently intense to be statistically detected after three years of experimentation, which has been shown before [16].

Changes in crop development and leaves nutritional status were also subtle. Alburquerque et al. [51] reported a similar increase in plant growth due to the fertilizing effects of compost made from olive and cotton gin wastes when comparing with control. Some studies showed significant improvements of tree growth [52], while others found no significant effect of the organic fertilization [53]. Baldi et al. [54] also found that peach trees' growth was not affected by fertilization strategy under field conditions. Ben Abdallah et al. [55] found no differences in vegetative development, which was explained by the fact that organic materials were not affecting the entire soil profile. All these results suggest that organic amendments for crop development are not always the most limiting factor, especially under rainfed conditions, being in our case the amount of rainfall probably a more decisive aspect.

The deficiency of N in leaves observed in all treatments (normal values ranged 1.4–2%) indicated the lack of N availability in organic and mineral treatments. Although leaf N values in rainfed crops are usually low, foliar applications of N should be considered in the future since the compost addition did not imply any advantage. It is worth noticing the apparent enriching effect on P and K that the compost had. However, differences between treatments in any of the other nutrients analyzed were not detected. It seems that three years was not a sufficient period to allow changes in the orchard's nutritional status to be detected, which is probably a consequence of the slow response by olive trees to changes in fertilization practices [34].

Crop productivity and harvest quality seemed to be more associated with the typical alternating phenomena of this particular tree, which greatly affects the harvests from year to year, rather that compost related. The results seem to indicate that, although olive trees have a natural resistance to drought, their growth, productivity and yield can greatly be undermined by the lack of water [55]. Some lifecycle assessments (LCAs) on olive crops have compared conventional to organic systems [56,57] including organic fertilization. In general, these studies agree that, although the environmental impacts of organic systems were lower than those of conventional systems, the productivity obtained with the organic systems should be improved to be competitive for the grower.

A positive effect on olive yield, as a consequence of the increase in organic matter content, total N and available P and K, was found by López-Piñeiro et al. [58] with olive waste used as soil amendment. However, their results were based on a period of 5 years of compost application and amendments were applied annually so that effects were probably more intense than the ones produced in this study with just two applications.

Overall, our results sustain that organic fertilization could be as efficient as mineral fertilizer to maintain not only quantity but also quality of table olives.

### 4.2. Potential Negative Effects of Compost Addition

Some of the most common environmental and agronomic drawbacks of compost addition include gaseous emissions and increase in salt and heavy metal contents [59].

In general, values of trace elements in vegetal tissues were within the ranges considered normal for plants [60]. Regarding Cu, a significant increase was observed during the second sampling (2019), but this fact is probably related to the application of Cu sulfate as a fungicide to the crop. Gascó and Lobo [34] also found no significant increases in trace elements in olive seedlings (more sensitive than adult trees) treated with sewage sludge. Authors who have found signs of toxicity in the plant have normally attributed them to salinity and sodium problems and not to heavy metals of the amendment. In our case, no visual toxicity symptoms were observed in any treatments and Na contents were always within the normal range for olive tree. These findings further reinforce the idea that these issues are directly linked with the quality of the final compost [16].

It is worth noticing that BC compost seemed to have a certain detrimental effect on production right after the first year of compost application. This effect, which could be related to the lack of maturity of the compost, disappeared in the following campaigns. These results agree with those obtained by Gonzalez et al. [50] in field experiments in which the biosolid (composted or not) were not positive either from the agronomic point of view (crop production) or considering the vegetative response of the olive trees. This effect did not seem to be so related to the heavy metal content present in the materials but was linked to the lack of maturity in the "composted" sludge. In fact, the authors estimated that much of the phytotoxicity could come from this immaturity of the compost.

The possible adverse effect due to the lack of maturity in the BC compost seemed to have disappeared. In the case of the low doses of compost, the residual effects were probably not enough to maintain yields comparable to the mineral fertilizer after more than one year of the application. In the third season, again with low production of olives, and after a new application of organic amendments, the yield was similar among treatments.

## 5. Conclusions

Although increases in organic matter in soils do not always imply an increase in their agronomic productivity, it is necessary to take a more global view of the advantages of this type of sustainable fertilization that promotes the circular economy and zero waste. Composting and use of compost from urban and agricultural wastes can be an environmentally friendly solution to the disposal problem of these wastes and an adequate low-cost strategy for their recycling. The results in regulation services (hydraulic properties and increased C sequestration) and support (plant nutrients and soil water content) show the gradually positive effects of using these amendments. In addition, the results in production and quality of the crops were quite positive when a residue from the olive grove was added such as alperujo compost. Our results support that organic fertilizers could be used as an alternative to inorganic amendments maintaining the quality and quantity of table olives. Perhaps the results are more debatable in the case of biosolid compost with regard to the entrapment services, but they point to improvements in the rest of the services and that they have long-term profitability. Future research should assess the effect of the compost in a scenario under less limiting factors, e.g., droughts, to avoid masking possible beneficial effects of the compost. In this sense, more work needs to be done to gain more efficient water use under rainfed conditions through compost addition.

**Supplementary Materials:** The following are available online at https://www.mdpi.com/article/10.3390/agronomy11061223/s1. Figure S1: Total monthly rainfall and monthly mean values of average temperature and relative humidity. The graph indicates the sampling time and compost application. Period: January 2018–August 2020. Table S1: Sequence and chronological steps in compost application and samplings. Table S2: Trace metals in soil under different treatments: AC (alperujo compost), AC+ (high dose of alperujo compost), BC (biosolid compost) and BC+ (high dose of biosolid compost). Table S3: Trace metals in leaves under different treatments: AC (alperujo compost), AC+ (high dose of alperujo compost), BC (biosolid compost) and BC+ (high dose of biosolid compost).

**Author Contributions:** Conceptualization, E.M., E.B.; methodology, E.M., E.B.; formal analysis, L.L.d.S., E.M., E.B.; investigation L.L.d.S., I.G., E.M., E.B.; data curation, L.L.d.S., I.G., E.M., E.B.; writing—original draft preparation, L.L.d.S., E.M.; All the author have visualized and reviewed the manuscript. All authors have read and agreed to the published version of the manuscript.

**Funding:** This study was funded by the Spanish Ministry of Science and Innovation (project–AGL2017-84745 ORESTES) and European FEDER funds.

**Institutional Review Board Statement:** Not applicable.

**Informed Consent Statement:** Not applicable.

**Data Availability Statement:** The data presented in this study are available in the article or Supplementary Material.

**Acknowledgments:** Lozano thanks the Junta Andalucía and European Union for the research grant awarded in the area of the Andalusian Research. Development and Innovation (PAIDI 2020). We thank Ana Ruiz for their help applying the amendments and tilling the soil. Pilar Burgos and the staff of the IRNAS Analytical Service carried out the chemical analyses of soil and plant samples. Cristina García de Arboleya and Patricia Puente helped in the field and laboratory.

**Conflicts of Interest:** The authors declare no conflict of interest.

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
