# Peer review of "Agro-Industrial and Urban Compost as an Alternative of Inorganic Fertilizers in Traditional Rainfed Olive Grove under Mediterranean Conditions"

_agronomy, doi:10.3390/agronomy11061223_

Round 1
Reviewer 1 Report
Dear editors of Agronomy,
Thank you for sharing the manuscript with me. My observations are summarized below:
- Could you please rewrite the abstract in a more layman style to make more understandable for a wider public? Please avoid the abbreviations but rather use the full names. In the title “inorganic” is mentioned while in the abstract solely “organic”. Is this correct? Please better structure the abstract to have there more about the method and data. Please more highlight the results and say a bit more about the key recommendations derived from your results.
- Introduction. I think that it doesn´t make sense to introduce the abbreviation at the end of the first sentence and not to use it further. I would propose to delete this abbreviation.
- The framing of the study urgently needs to be expanded as is too general and not so the research problem-focused. It might be good if several subsections of the Introduction are developed that are dealing the studied issues individually. One of the sections could be focused solely on Mediterranean countries and say more about the context (please include the numbers so that we can see the scale of the problem).
- The methodology section seems to be well developed, however, the authors should be clearer when talking about the sequence of the steps conducted. A graphical scheme showing this would be good here. Please add the numbers to individual formulas and careful explain all the variables.
- Results look reasonable and interesting to me. It might be good if more graphics is used to present the findings. This would surely add more to the level of attractiveness and visibility of the paper.
- Please check the references to tables and figure if these are correct.
- The list of references seems to be written in wrong format, please consult the guide. It also seem to me that the list of refs is a bit outdated, please update for the most recent studies.
- Please develop the section on discussion independently (not as a part of the results section).
- Conclusion needs to be expanded and better show what has been found both empirically and conceptually. Are there any advances? Please add more about the limitations that could affect the replicability of your study.
I think that this is going to be a nice paper, however, some work still has to be done. Please focus on the theoretical framing of the study and the methods section.
I recommend a major revision.
Kind regards,
Author Response
Manuscript ID.: agronomy-1246735
Title: Agro-industrial and urban compost as an alternative of inorganic fertilizers in traditional rainfed olive grove under Mediterranean conditions
Authors: L. de Sosa et al.
We thank the editor and both anonymous reviewers for their supportive and very helpful comments on our manuscript. We have taken on board all the suggestions and modified the manuscript accordingly. You will find below a detailed reply to each of the reviewer’s comments. We feel that clarification of the points raised has greatly helped its readability. If further clarification is required on any points, we would be happy to do that.
Main amendments include:
(i) Firstly, as requested by Reviewer #1 introduction offer a more complete framework of the study.
(ii) We have also separated results and discussion as requested by the two reviewers.
(iii) Finally, as requested by Reviewer #1, references have been checked and a professional language editing service has revised the whole manuscript. New references more up to date have also been included.
Further detailed responses to points raised by each Reviewer are provided below. Thank you for considering our paper for publication in Environmental Microbiology; we consider the manuscript is now much improved and appreciate the opportunity of re-submitting
Reviewer 1.
- Could you please rewrite the abstract in a more layman style to make more understandable for a wider public? Please avoid the abbreviations but rather use the full names. In the title “inorganic” is mentioned while in the abstract solely “organic”. Is this correct? Please better structure the abstract to have there more about the method and data. Please more highlight the results and say a bit more about the key recommendations derived from your results.
Response: As suggested, the abstract has carefully revised providing more information about methods and data as well as more recommendations.
- Introduction. I think that it doesn´t make sense to introduce the abbreviation at the end of the first sentence and not to use it further. I would propose to delete this abbreviation.
Response: Thank you for raising this issue. The abbreviation has now been deleted.
- The framing of the study urgently needs to be expanded as is too general and not so the research problem-focused. It might be good if several subsections of the Introduction are developed that are dealing the studied issues individually. One of the sections could be focused solely on Mediterranean countries and say more about the context (please include the numbers so that we can see the scale of the problem).
Response: As suggested introduction has now been complemented with a more detailed Mediterranean context.
- The methodology section seems to be well developed, however, the authors should be clearer when talking about the sequence of the steps conducted. A graphical scheme showing this would be good here. Please add the numbers to individual formulas and careful explain all the variables.
Response: Thanks for the suggestion. New material has been provided in the supplementary in order to provide the sequence of the experiment (Table S1).
- Results look reasonable and interesting to me. It might be good if more graphics is used to present the findings. This would surely add more to the level of attractiveness and visibility of the paper.
Response: We completely agree that more graphics would increase the level of attractiveness and visibility of the manuscript. Unfortunately, we just left the data with very few variations in their figures without plotting. We honestly think that this data will not add any value to the manuscript but changes could be made accordingly if necessary
- Please check the references to tables and figure if these are correct.
Response: Thank you. References have been fully checked.
- The list of references seems to be written in wrong format, please consult the guide. It also seems to me that the list of refs is a bit outdated, please update for the most recent studies.
Response: References matched the format provided by agronomy in their templates. New references have also been added.
- Please develop the section on discussion independently (not as a part of the results section).
Response: As suggested, the results and discussion section are now independent.
- Conclusion needs to be expanded and better show what has been found both empirically and conceptually. Are there any advances? Please add more about the limitations that could affect the replicability of your study.
Response: Conclusion has been expanded adding more of the findings we got.
Reviewer 2 Report
During the cultivation of the plants, was examined and then the ANOVA and post-hoc tests were performed. The methodology of the statistical research has not been properly described (line 190-187). There is probably more than ANOVA and the limit of 0.05 (algoritm, program, etc). See how it was done in this article and refer to it.
Roman, K.; Roman, M.; Szadkowska, D.; Szadkowski, J.; Grzegorzewska, E. Evaluation of Physical and Chemical Parameters According to Energetic Willow (Salix viminalis L.) Cultivation. Energies 2021, 14, 2968. https://doi.org/10.3390/en14102968
The resignation and discussion should be separated, because the discussion should include references to other studies.
Are the graphs in Figure 2 really good to present? Should the line connecting the totals go there? Do you know what happens at <psi> 100? is the waveform surely linear?
Author Response
Response to the reviewer comments
Manuscript ID.: agronomy-1246735
Title: Agro-industrial and urban compost as an alternative of inor-ganic fertilizers in traditional rainfed olive grove under Medi-terranean conditions
Authors: L. de Sosa et al.
We thank the editor and both anonymous reviewers for their supportive and very helpful comments on our manuscript. We have taken on board all the suggestions and modified the manuscript accordingly. You will find below a detailed reply to each of the reviewer’s comments. We feel that clarification of the points raised has greatly helped its readability. If further clarification is required on any points, we would be happy to do that.
Main amendments include:
(i) Firstly, as requested by Reviewer #1 introduction offer a more complete framework of the study.
(ii) We have also separated results and discussion as requested by the two reviewers.
(iii) Finally, as requested by Reviewer #1, references have been checked and a professional language editing service has revised the whole manuscript. New references more up to date have also been included.
Further detailed responses to points raised by each Reviewer are provided below. Thank you for considering our paper for publication in Environmental Microbiology; we consider the manuscript is now much improved and appreciate the opportunity of re-submitting
Reviewer 2
During the cultivation of the plants, was examined and then the ANOVA and post-hoc tests were performed. The methodology of the statistical research has not been properly described (line 190-187). There is probably more than ANOVA and the limit of 0.05 (algoritm, program, etc). See how it was done in this article and refer to it.
Response: As suggested, we have included a brief explanation of the statistical analysis referring to the article provided.
Roman, K.; Roman, M.; Szadkowska, D.; Szadkowski, J.; Grzegorzewska, E. Evaluation of Physical and Chemical Parameters According to Energetic Willow (Salix viminalis L.) Cultivation. Energies 2021, 14, 2968. https://doi.org/10.3390/en14102968
The resignation and discussion should be separated, because the discussion should include references to other studies.
Response: As suggested, the results and discussion section are now independent.
Are the graphs in Figure 2 really good to present? Should the line connecting the totals go there? Do you know what happens at <psi> 100? is the waveform surely linear?
Response: We consider that figure 2 add value to the manuscript as some differences in some of the soil hydraulic properties were identified. We agree that the waveform could vary at <psi> 100 being not completely linear. However, the discussion is focused on the particular points of pressure we measured and judging by the other results it doesn’t seem the trend would go to change abruptly.
Round 2
Reviewer 1 Report
I have no further comments. The authors sufficiently reflected or commented on all my observations. Thank you. Kind regards,Author Response
Dear Reviewer,
Thank you for your suggestions and recommendations. The spelling has been checked again and we feel it has greatly helped its readability.
Kind Regards,
Laura Lozano
Reviewer 2 Report
The article has been corrected and I have no comments. The statistics seem clear and the analysis of homogeneous groups introduced is correct. Moderate English changes required.
Author Response
Dear Reviewer,
Thank you for your suggestions and recommendations. The spelling has been checked again and we feel it has greatly helped its readability.
Kind Regards,
Laura Lozano